# An Estimated Structural Equation Model to Assess the Effects of Land Use on Water Quality and Benthic Macroinvertebrates in Streams of the Nam-Han River System, South Korea

**DOI:** 10.3390/ijerph17062116

**Published:** 2020-03-23

**Authors:** Jong-Won Lee, Sang-Woo Lee, Kyung-Jin An, Soon-Jin Hwang, Nan-Young Kim

**Affiliations:** 1Graduate Program, Department of Forestry and Landscape Architecture, Konkuk University, Gwangjin-Gu, Seoul 05029, Korea; jwlee8901@konkuk.ac.kr; 2Department of Forestry and Landscape Architecture, Konkuk University, Gwangjin-Gu, Seoul 05029, Korea; dorian@konkuk.ac.kr; 3Department of Environmental Health Science, Konkuk University, Gwangjin-Gu, Seoul 05029, Korea; sjhwang@konkuk.ac.kr (S.-J.H.); celeste0@hanmail.net (N.-Y.K.)

**Keywords:** land use, water quality, stream, structural equation model, benthic macroinvertebrates

## Abstract

The extent of anthropogenic land use in watersheds determines the amount of pollutants discharged to streams. This indirectly and directly affects stream water quality and biological health. Most studies have therefore focused on ways to reduce non-point pollution sources to streams from the surrounding land use in watersheds. However, the mechanistic pathways between land use and the deterioration of stream water quality and biological assemblages remain unclear. This study estimated a structural equation model (SEM) representing the impact of agricultural and urban land use on water quality and the benthic macroinvertebrate index (BMI) using IBM AMOS in the Nam-Han river systems, South Korea. The estimated SEM showed that the percent of urban and agricultural land in the watersheds significantly affected both the water quality and the BMI of the streams. Specifically, a higher percent of urban land use had directly increased the biochemical oxygen demand (BOD) and total phosphorus (TP), and deteriorated the BMI of streams. Similarly, higher proportions of agricultural land use had also directly increased the BOD, total nitrogen (TN), and total phosphorus (TP) concentrations, and lowered the BMI of streams. In addition, it was observed that the percent of urban and agricultural land use had indirectly deteriorated the BMI through increased BOD. However, we were not able to observe any significant indirect effect of the percent of urban and agricultural land use through increased nutrients including TN and TP. These results indicate that increased urban and agricultural land use in the watersheds had directly and indirectly affected the physicochemical characteristics and benthic macroinvertebrate communities in streams. Our findings emphasize the need to develop more elaborate environmental management and restoration strategies to improve the water quality and biological status of streams.

## 1. Introduction

Land use affects the physical and chemical characteristics of streams, which significantly impacts water quality and fluvial ecosystems [1,2,3]. Anthropogenic activities influence the type, intensity, and amount of land use within a watershed and have been closely associated with the amount of material and nutrients discharged into nearby streams. The proportion of anthropogenic land use in watersheds can therefore be a key indicator of stream water quality [4,5,6]. According to previous studies, high proportions of urban and agricultural areas in watersheds are linked to high stream organic matter and nutrient concentrations [7,8,9]. Specifically, urban land use alters the hydrological regime in watersheds [10,11] and increases the outflow of both point source and non-point source pollution from roads, commercial facilities, and residential areas [7,12,13]. Nutrients from agricultural land are a dominant non-point pollution source to streams [14] due to the excessive application of fertilizers [15] and pesticides [16] for enhanced agricultural productivity.

Land use also significantly impacts the biological community structure of streams [17,18,19,20,21]. For example, the rapid urbanization in the watersheds of Jialing and Yangtze Rivers in China predominantly contributed toward the degradation of aquatic ecosystems in streams [22]. In particular, the population of pollution-tolerant species (Tubificidae, Chironmidae, Belostomatidae, Physidae, and Glossiphoniidae) from urban areas significantly increased, which negatively influenced the macroinvertebrate community structure. Low species richness and the absence of intolerant benthic macroinvertebrate species are commonly observed in urban streams [19]. Pollution from agricultural land had also deteriorated macroinvertebrate habitats and negatively impacted the benthic macroinvertebrate community in a stream of the Nam-Han River in Songcheon, South Korea [23]. Further, pollutants generated in urban and agricultural areas have been reported to adversely affect water quality and the structure of biological communities [24]. For instance, sediments from agricultural lands smoothens the river bed, making it less rugged, and results in an increased number of macroinvertebrate communities that are tolerant to habitat quality [25,26]. Agricultural land use involves removing riparian vegetation and installing weirs, which reduces the flow rate and causes sediments to build up in the stream bed and smoothens the stream bed substrates. Streams in the intensively cultivated agricultural areas in Oregon, USA, were dominated by macroinvertebrate communities that were tolerant to organic pollution, had a low level of dissolved oxygen, and showed low species diversity [27]. In the urban areas of the Iberian peninsula, physical and chemical stress including metals, nutrients, and temperature reduced the number of macroinvertebrate communities [28]. 

To analyze the links between land use, water quality, and aquatic ecosystems in watersheds, previous studies have predominantly applied the ordinary least squares (OLS) approach, which mainly incorporates gradient techniques based on linear regressions, such as multiple regression models and canonical correlation analysis [29]. This statistical approach can explain the changes in physiochemical characteristics and biological communities of streams through linear correlations with a number of variables [29]. However, OLS regression is limited because it only assesses linear relationships and cannot adequately capture the complexity of ecological datasets [29,30,31]. Many explanatory variables in streams are at risk of multicollinearity. This suggests that time and space interactions can cause spurious correlations between variables in a multivariate dataset within a given spatial unit [29,30,31]. Structural equation modeling (SEM) is an extension of multiple regression analysis and was developed to quantify the complex relationships between such variables; it statistically assesses the structured causal relationships between multiple dependent and independent variables [32]. SEM is a useful statistical technique for determining the complex causality between land use, water quality, and aquatic organisms. While linear regression assesses only the direct impacts of environmental variables on aquatic organisms, SEM assesses both the direct and indirect impacts of the dependent variables on the dependent variable. These cause-effect relationships are inferred based on statistical correlations presumed to represent ecologically meaningful pathways [29]. As such, the model attempts to interpret the covariance or correlation between variables. Further, the maximum likelihood estimation (MLE) is generally applied in the SEM. Based on this rationale, the causality and correlation between variables can be modified to fit the model.

This approach has been widely used in ecological studies because it accounts for several aspects of biological variations over space with one or several variables affecting river conditions [29,33]. In a previous study, SEM was effectively used to investigate the indirect effects of agricultural land use on the biological integrity in eight regions of the United States [33]. In the study, agricultural land use in watersheds affected water quality, causing eutrophication and affecting macroinvertebrate communities. In addition, SEM was used to gain insight on the factors affecting river conditions by evaluating the direct and indirect effects of human system, terrestrial landscape, and physical stream variables on stream invertebrates [33]. 

In this study, we aimed to develop a SEM to assess the relationships between land use, water quality parameters, and benthic macroinvertebrates indexes in streams of the Nam-Han River watershed, South Korea. The majority of studies have predominantly focused on one-dimensional relationships, such as changes in the health of aquatic ecosystems in response to altered land use. This type of approach can determine the dominant causes (i.e., land use around stream) and effects (i.e., health deterioration of aquatic ecosystem) but cannot identify the structural relationships between them. Based on previous studies, this study hypothesized that the urban and agricultural land use proportions would directly impact the status of benthic macroinvertebrate communities in streams and indirectly affect benthic macroinvertebrate communities, through stream physicochemical water quality (i.e., BOD, TN, and TP). The results of this study elucidate the complex relationships between land use, water quality, and aquatic ecosystems, which are essential for improving environmental management and restoration practices in polluted watersheds. Better understanding of the true nature of the complex relationships between land use, water quality, and aquatic ecosystems is expected to provide profound insights on the more effective watershed management and stream restoration strategies.

## 2. Materials and Methods

### 2.1. Study Area

Our study area was the Nam-Han River watershed located in the center of the Korean peninsula (Figure 1). The Nam-Han River has a length of 375 km and covers approximately 37% (12,577 km^2^) of the total area of the Han River (34,473 km^2^). The Nam-Han River watershed consists of alpine topography, reaching elevations of >1000 m. The river flows in the form of an incised meander at midstream elevations of approximately 200m to 500 m, settling at 100 m in the downstream plain [34]. The average annual rainfall in the watershed is approximately 1409 mm, which exceeds the national average of 1159 mm. Two-thirds of the total annual precipitation occurs in summer (June-September), and streams dry out during spring (March–May) and fall (October–November), particularly in monsoon weather. The long-term annual average temperature over the last 30 years is 24.08 °C, with the lowest and highest average monthly temperatures in January (−3.9 °C) and August (24.4 °C), respectively [35].

The watershed largely consists of forests, covering approximately 70% of the total area. Agricultural and urban areas are concentrated downstream and cover approximately 20% and 4% of the total area, respectively. The urban areas significantly impact river water quality, despite their low coverage in the watershed [17]. Of particular concern are the high concentrations of harmful compounds such as phosphorus and nitrogen sourced from livestock wastewater and agricultural activity in the surrounding watershed. These compounds cause physicochemical pollution and promote stream eutrophication. The Nam-Han River accounts for 54% of the total inflow to the Paldang reservoir—a major water source to the metropolitan area [24].

### 2.2. Sampling Sites

The Ministry of Environment (MOE) in Korea along with the National Aquatic Ecological Monitoring Program (NAEMP) developed evaluation standards and sampling protocols to monitor the river from 2003 to 2006. Since 2007, the MOE has been monitoring the long-term changes of a number of ecological and environmental indices (including the Trophic Diatom Index (TDI), the Benthic Macroinvertebrate Index (BMI), the Fish Assessment Index (FAI), the Habitat and Riparian Index (HRI), the Riparian Vegetation Index (RVI), and the physical and chemical water quality index) at 800 monitoring sites along 5 major rivers. The MOE expanded the number of sampling sites to 3,039 in 2019 to increase the sampling spatial resolution. In this study, we used the 2018 datasets from 111 sampling sites in the Nam-Han River watershed (Figure 1).

### 2.3. Water Quality

NAEMP measured various water quality parameters in each sampling site, including biochemical oxygen demand (BOD), ammonia/ammonium (NH_3_-N), nitrate (NO_3_-N), total nitrogen (TN), total phosphorus (TP), phosphate (PO_4_-P), and chlorophyll-a. Previous studies on land use and aquatic organisms applied various water quality parameters in their analysis. However, in this study, we only selected organic pollutants and nutrient variables that are known to be sensitive to land use (e.g., [18,36]). In particular, organic pollution frequently occurs in streams and lakes. BOD is generally applied in Korea and is useful for water quality management policies as an indicator of organic pollution [37]. Anthropogenic N and P concentrations are also known to significantly impact eutrophication and water quality in streams [38]. We therefore selected BOD as the organic pollution indicator and TN and TP as the eutrophication indicators in this study.

### 2.4. Benthic Macroinvertebrate Index

Benthic macroinvertebrates are useful biological tools for assessing the pollution status of aquatic ecosystems and have been commonly applied in various countries over the years [39]. Benthic macroinvertebrates are relatively easy to collect and identify, and different species react differently to stream disturbances [40]. They play significant roles in aquatic ecosystem functioning, as they are both primary and secondary consumers and are an important food source for fish [41]. Benthic macroinvertebrates also respond rapidly to environmental (substrate, depth, water velocity, water temperature, etc. [42]) and physicochemical factors [30,43] and are highly sensitive to minor changes in stream and watershed environments [44]. As such, they are ideal for evaluating the environmental status of streams [26,45] in the context of both water quality and ecological health [46].

To assess stream health, we calculated the BMI (Equation (1)) at each site using indicator species from a total of 190 classifications. In Korea, the BMI is calculated based on the appearance, indicator weight value, and saprobic value of species in accordance with the method used by Zelinka [47]:(1)$$\mathrm{BMI}=\left(4-\frac{\sum_{i=1}^{n}s_{i}h_{i}q_{i}}{\sum_{i=1}^{n}h_{i}q_{i}}\right)\times 25$$
where i is the number assigned to the species, n is the number of species, $s_{i}$ is the unit saprobic value of species i, $h_{i}$ is the frequency of species i, and $q_{i}$ is the indicator weight value of species i. The BMI is scored from 0 to 100, with 100 indicative of excellent aquatic ecosystem health and 0 indicative of poor aquatic ecosystem health [46].

### 2.5. Land Use/Land Cover

In 1998, the MOE produced land cover maps of the Korean Peninsula using Landsat TM satellite imagery to promote environmentally friendly land use and management practices in the region. Land cover was a scale of 1:50,000 and was classified into seven categories: urban, agricultural, grassland, wetland, bare land, and water. In 2000, the MOE produced another land cover map based on scale of 1:25,000. In 2006, the map was further upgraded using satellite imagery from Arirang 2, resulting in a total of 22 land cover classifications [48]. These land cover maps are commonly used for calculating non-point source pollution loads, biotope mapping for urban planning, and evaluating environmental impacts. Governments also use these land cover maps to establish environmental policies. The MOE classified the Nam-Han River watershed into 8 mid-watersheds and 103 sub-watersheds for water environmental management. We adopted the sub-watershed as the spatial unit in this study. We used a land cover map provided by the Environmental Geographic Information Service (EGIS) to determine the land use proportions near the stream. The land cover map has a spatial resolution of 5 m, and a 1:25,000 topographic map was used for the Arc GIS 10.4 program analysis. Land use in the watershed, urban areas, and agricultural areas were extracted from the land cover map based on the seven classifications. The urban area includes residential, industrial, commercial, cultural/physical education/leisure, transportation, and public facility areas. The agricultural area includes farmland, cultivation facilities, orchard, and other cultivation areas (Figure 2).

### 2.6. Model Estimation

To verify normality, the Kolmogorov-Smirnov test was conducted and the AMOS 18 software of SPSS was applied to estimate the causal relationship between variables; this was based on the knowledge that higher proportions of urban and agricultural land in watersheds negatively impact the stream environment by organic and nutrient accumulation [18,36,49]. In turn, water quality deterioration negatively impacts the benthic macroinvertebrate community structure [22,23]. Based on this research model, we analyzed the structural correlations between urban/ agricultural land use and BMI in the Nam-Han River watershed, with water quality as the mediator (Figure 3). Figure 3 illustrates the causal relationships between urban and agricultural land use, water quality (BOD, TN, TP), and BMI, demonstrating that both land use types directly and indirectly affect BMI. D1–D4 represent structural errors, which means they were excluded in the model, and thus the variance cannot be explained by these variables. This infers the potential influence of other variables that are not included in the path model in Figure 3. 

The relationship between variables in SEM is expressed by the same path coefficient as the standardized coefficient. SEM estimates the path coefficient and enables measurements of direct, indirect, and total effects. Information on causal mechanisms are obtained by comparing these effects. Furthermore, the model provides a simple interpretation on the relationships between variables by decomposing correlation coefficients. It also identifies the relative sizes of direct and indirect effects of one variable over another. Finally, the frequency of change in outcome variables can be determined depending on the alternation of antecedent variables through the total effects [50,51]. Direct effects are the direct impacts of one variable on another. Indirect effects are effects that occur through other variables, which are mediated by the intervention between exogenous and endogenous variables under the total effects, i.e., the effect of other variables between antecedent and outcome variables [50]. To identify the amount of influence of each variable, the model therefore decomposes the correlation among variables into direct, indirect, and total effects [32]. In this study, the model decomposes the effects of urban and agricultural land use, which in turn affects the SEM correlation between variables. The size of the influence for each variable is then estimated. To test the statistical significance of the estimated parameter in SEM, the Critical Ratio (C.R.) is used. The C.R. is the ratio of any one deviation from the mean in a set of observed values of the same statistical variable to the standard deviation of the set or to the corresponding probable error. The C.R. based on a significance level of 0.05, needs to be > ± 1.96 [52].

### 2.7. Model Validation Criteria

The most appropriate goodness-of-fit index to evaluate the model suitability has not yet been determined. Therefore, it is important to have an appropriate standard and to select an appropriate fit index for individual model evaluations [53]. The most commonly applied method to identify goodness-of-fit is the Chi-squared verification. However, its use has been limited in recent years due to its sensitivity to sample size: i.e., for the same model size, the results can be valid or invalid depending on the size of the sample [53,54]. More recent studies have instead shown preference toward fit indexes, including the normed fit index (NFI), the Tucker-Lewis Index (TLI), the Comparative Fit Index (CFI), the goodness-of-fit (GFI), the adjusted goodness-of-fit index (AGFI), and the root mean square error of approximation (RMSEA) [53].

Relative fit indexes include NFI, TLI, and CFI; they indicate how well the research model compares with the lowest performing model. TLI is one of the oldest relative fit indexes and is not affected by sample size, resulting in minimized model error. The TLI value ranges from 0 to 1 (sometimes > 1), and the value increases if it is simple and clear. In general, values >0.9 indicate a good fit. NFI was developed to prevent TLI values exceeding 1. Higher NFI values indicate a stronger model fit, particularly at values >0.9. CFI was developed to minimize the effect of sample size on NFI. It is therefore ideal for evaluating the goodness-of-fit of the model, as it is not influenced by sample size. A CFI of >0.9 indicates a good model fit. Absolute Fit Indexes (AFI) include GFI, AGFI, and RMSEA; these indexes provide absolute evaluation by comparing their suitability with other models. GFI shows how well the research model compares with the null model, but it is affected by sample size and does not consider model simplicity. AGFI was therefore developed to improve upon these limitations; however, it is also affected by sample size and is less frequently applied. Values of >0.9 and >0.85 indicate good model fitting for GFI and AGFI, respectively. RMSEA is not affected by sample size and also considers its error and simplicity. The model indicates a good fit if values are <0.05, a mediocre fit if values are <0.1, and an unacceptable fit if values are >0.1 [53,55]. In this study, we applied NFI, TLI, CFI, GFI, AGFI, and RMSEA to evaluate the suitability of the model [53].

## 3. Results

### 3.1. Statistical Analysis

#### Descriptive Statistics

All variables in this analysis were not normally distributed and were therefore log-transformed. We also calculated descriptive statistics to analyze the variables, including those that were log-transformed (Table 1). On the sub-watershed scale, urban and agricultural land covered an average area of 5.87% and 26.75% in the Nam-Han River watershed, respectively. The coverage of agricultural land was therefore significantly higher than urban land. The maximum (62.50%) agricultural area was concentrated in the lower part of the stream. BOD concentrations ranged from 0.50–7.60 mg/L, with an average of 1.81 mg/L; TN concentrations ranged from 0.69–11.20 mg/L, with an average of 3.85 mg/L; and TP concentrations ranged from 0.00–0.39 mg/L, with an average of 0.05 mg/L. BOD and TP levels were classified as “very good” and “slightly good”, respectively, based on Article 2 of “Act on Environmental Policy” enforcement ordinance in South Korea. BMI ranged from 22.6–94.7, with an average value of 66.71 (Table 1).

### 3.2. Estimated SEM

#### 3.2.1. Direct and Indirect Effects from the SEM

The urban and agricultural land use directly increase the concentration of BOD. The direct effects of urban and agricultural land use on TN were *β* = 0.07 and *β* = 0.39, respectively, but the impact of urban land use was not significant. The direct effects of urban and agricultural land use on TP were *β* = 0.24 and *β* = 0.45 respectively, and their direct effects on BMI were *β* = −0.30 and *β* = −0.21, respectively. The BOD, TN, and TP directly decreased the concentration of BMI, but the impacts of TN and TP concentrations were not significant. The indirect effects of urban and agricultural land use on BMI through BOD were *β* = −0.06 and *β* = −0.21 but their effects were not significant (Figure 4).

#### 3.2.2. Refined Model

We modified the estimated SEM by eliminating insignificant pathways of low TLI values. Urban land use, agricultural land use, and BOD concentration significantly affected BMI based on the results of the SEM analysis. In contrast, TN and TP showed no significant effects on BMI. Further, the direct effects of urban land use on TN were not significant and were therefore removed and corrected (Urban → TN, TN → BMI, TP → BMI). After deleting insignificant pathways, the significant standard regression weight of the estimated path model and the modified model showed no significant difference. Moreover, the Urban → TP, Agricultural → BMI, and BOD → BMI pathways became more meaningful in the refined model (Figure 5).

We conducted a model validation based on multiple goodness-of-fit criteria for the modified SEM.. According to the goodness-of-fit criteria, we conclude that the refined model is suitable for explaining the data in this study (Table 2).

### 3.3. Relationships Between Variables in the Refined Model

#### Relationships Between Urban/Agricultural Land Use, Water Quality, and BMI

The percent of agricultural land use in the watershed shows statistically significant path coefficients for BOD, TN, and TP, suggesting that the water quality concentration increases with the proportion of agricultural land use. The urban land use proportion shows statistically significant path coefficients for BOD and TP, suggesting that the water quality concentration increases with the urban land use proportion. In addition, the higher the urban and agricultural land use proportion, the lower the BMI; and the higher the BOD concentration, the lower the BMI (Table 3).

## 4. Discussion

### 4.1. Impact of Land Use on Water Quality

The proportion of agricultural and urban land use in watersheds is a major determinant of BOD, TN, and TP in streams [18] and has been frequently linked to nutrient and heavy metal contamination from non-point pollution sources, such as pesticides and fertilizers [56,57]. However, we did not observe a significant relationship between the proportion of urban areas and TN in this study. In general, the high number of non-point nitrogen sources in urban areas exacerbates stream pollution, and the nitrogen loss is typically higher than that of agricultural areas [58]. The research sites in this study are spatially concentrated in the urban and agriculture regions downstream of the Nam-Han River. Despite the low coverage of urban areas relative to that of agriculture, urban land was shown to significantly impact the water quality of nearby streams. BOD and nutrient concentrations may therefore be affected depending on the proportion or range of urban areas in the watershed [57].

In this study, agricultural areas negatively affected all water quality indicators, which is consistent with previous findings [5,21]. Pesticides, fertilizers, compost, and manure from agricultural land—such as paddy areas and hoop houses—are discharged in runoff to nearby streams [56]. Excretion and gas emissions from agricultural land also exacerbate water quality [25]. The agricultural areas in the Nam-Han River watershed consist of 41.36% paddy areas and 53.23% dry-lands, the majority of which (97.83% of paddy areas and 75.94% of dry-lands) are located downstream of the Nam-Han River, in an area with a high proportion of livestock. According to the Water-Environmental Management Plan (2017–2021) of the downstream Nam-Han River, the highest BOD pollutant load in 2015 was 32,555 kg/day, 60.3% (19,630 kg/day) of which was sourced from livestock pollutants. The most common point pollution source was poultry followed by livestock, while non-point pollution sources accounted for most of the pollution in streams. In general, livestock excretion accounts for only 0.6% of the total sewage and wastewater discharge; however, it contributes to 25.8% of the total pollutant load to streams due to the high nutrient levels in manure (N, P) [59]. In particular, nitrogen predominantly originates from non-point sources [60] and is mostly discharged into ground water, while phosphorus predominantly remains in surface discharge [59].

### 4.2. Impact of Land Use and Water Quality on Benthic Invertebrates

Nutrient concentrations were shown to negatively affect benthic macroinvertebrates in the Lower Grand River Watershed in the United States [5] and in urbanized areas of the Yangtze River in China [22]. A previous study that assessed the combined effects of flow, sediment, and nutrients on benthic macroinvertebrate communities in Danish rivers had demonstrated insignificant nutrient impact on benthic macroinvertebrates [61]. However, we were not able to observe the direct effect of nutrient on BMI. This insignificant relationship between nutrient and benthic macroinvertebrates has been reported in previous studies. For example, Kang [62] reported that phosphorus and nitrogen enrichment did not significantly impact benthic macroinvertebrate stressors in the Kildavin River watershed in south-east Ireland [62]. However, he reported the significant effects of nutrients on the benthic macroinvertebrate under high sediment loads. Thus, the effects of the nutrients on the benthic macroinvertebrate might vary, depending the conditions of other stream environments (i.e., sediments, physical environment, and biochemical conditions). For instance, a multi-spatial scale analysis of urbanization and benthic macroinvertebrates in China demonstrated that the composition of benthic macroinvertebrate communities varied according to habitat size, water flow, and riverbed temperature [22]. In addition, agricultural practices commonly decrease the deposition of sediments in streams. As benthic macroinvertebrates inhabit the riverbed, variations in the deposition of stream substrates can significantly impact their biodiversity. Stream substrates form the habitats of benthic macroinvertebrates, and thus, their size is dependent on the physical habitat environment [63]. Sedimentary deposits in the riverbed reduce the habitat quality and impair the breathing of benthic macroinvertebrates. In particular, the proportion of Ephemeroptera, Plecoptera, and Tricoptera are reduced due to their sensitivity to high sediment contamination. Although the results of this study showed insignificant impacts of the nutrient on the benthic macroinvertebrates, we are reluctant to generalize these findings in other regions due to the contrasting findings of previous studies. The concentration of nutrients in streams can indirectly affect benthic macroinvertebrates through their impact on the food supply [61]. Thus, the absence of a nutrient effect on benthic macroinvertebrates may be due to benthic algae, which are a dominant food source for benthic macroinvertebrates and are unaffected by nutrient loads [61]. However, this indirect effect of nutrient on benthic macroinvertebrates was not examined in this study due to lack of data. 

Oxygen reduction in response to high organic loading was found to affect benthic macroinvertebrates [64]. Persistent nutrient influx also reduces dissolved oxygen in water and may have toxic effects on benthic macroinvertebrates. In addition, increased BOD concentrations from nearby settlements and industrial facilities reduces the dissolved oxygen and water quality of streams; this negatively impacts aquatic fauna and flora, including benthic macroinvertebrates [64,65]. High BOD therefore reduces the number of species that are intolerant to low dissolved oxygen conditions, while species tolerant to low oxygen become more abundant. However, the abundance of species sensitive to nutrients therefore decreases [21], as they favor higher oxygen concentrations [66]. The results of this study confirm the findings of previous studies. We found BOD to be an important water quality variable that regulates the health of benthic macroinvertebrates. In general, urban land use accounts for a low proportion of the total watershed area, but wastewater and domestic sewage emissions from urban/industrial activities and dense settlements have a significant impact on river water quality and benthic macroinvertebrates [11,17]. Jabbar and Grote [5] found that water quality parameters, including nitrogen and phosphorus, were more closely correlated with urban areas as opposed to agricultural areas, as most of the effluents from urban sewage treatment plants are discharged to stream tributaries, causing a rise in BOD [25]. In this study, however, agricultural areas were more closely linked to stream BOD relative to urban areas. The urban, agricultural, and forested areas in the study by Jabbar and Grote [5] covered 4.6%, 76.1%, and 12.4% of the total basin area, respectively. In this study, agricultural coverage was much lower at 26.7%, while urban coverage was similar at 5.8%.

Considering the complex interactions among watershed characteristics, stream environments, and biological communities, the direct impacts of land use are difficult to identify from indirect impacts. Practically, the direct impacts of urban and agricultural land use might be associated with reducing or degrading habitat size and quality, resulting in a decreasing population size and varying species composition of benthic macroinvertebrates when developing urban and agricultural areas [67,68]. In particular, frequent construction projects (e.g., channelizing, removing flood plain, changing stream bed material, armoring stream bank, and removing riparian vegetation) near or in streams might be directly associated with rapidly degrading habitat size and quality, as well as poor biological condition of streams [69,70,71,72]. In particular, riparian vegetation is critical for sustaining the integrity of biological communities in streams; further, the positive effects of riparian vegetation on water quality and biological conditions have been well documented. Macroinvertebrate indicators were closely related to riparian vegetation structures, suggesting that the stream organisms provided local responses [73]. In contrast, the indirect effect of urban and agricultural areas on macroinvertebrate through BOD is related to the composition of the macroinvertebrate communities. Species susceptible to poor water quality are gradually decreasing, while tolerant species are gradually increasing, resulting in streams dominated by tolerant species [74]. 

## 5. Conclusions

The water quality of streams and the health of aquatic organisms are significantly affected by anthropogenic land use. In this study, we found urban and agricultural land use in the Nam-Han River watershed to significantly impact the water quality and benthic macroinvertebrate community (BMI) of nearby streams. Developed areas had negatively affected stream BOD, TN, and TP concentrations. Agricultural land use more strongly influenced water quality, while urban land use more strongly influenced BMI. In contrast to previous studies, benthic macroinvertebrate communities showed no significant relationship with nutrient load (TP and TN), but were significantly correlated with BOD.

According to our estimated SEM, urban land use had directly affected water quality (BOD, TP) and indirectly affected benthic macroinvertebrates through its impact on BOD. Agricultural land use had indirectly affected all water quality variables (BOD, TN, TP) and indirectly affected benthic macroinvertebrates through its impact on BOD. We therefore conclude that agricultural land use significantly impacts the chemical water quality index of streams. The model also sheds light on the relative influence of each variable on benthic macroinvertebrates. Accordingly, watershed managers can help to prioritize more sophisticated stream management and non-point pollution management to preserve and promote stream aquatic health.

## Figures and Tables

**Figure 1 ijerph-17-02116-f001:**
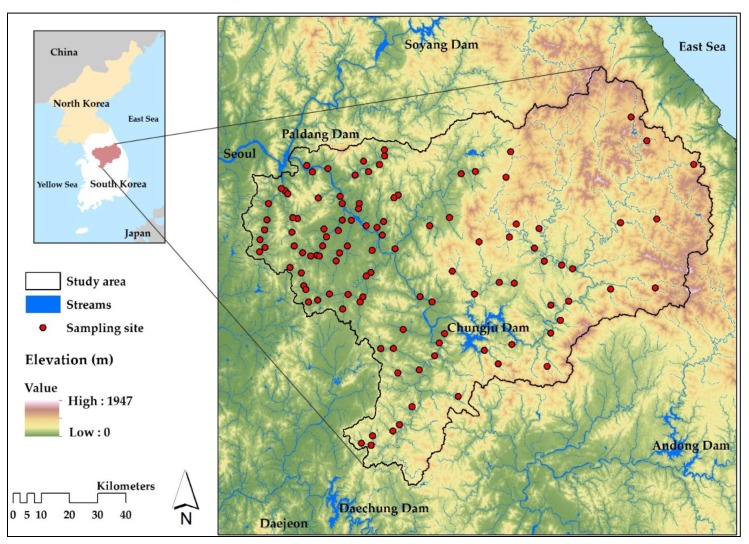
Topography the Nam-Han River watershed, including the locations of the 111 sampling sites from the National Aquatic Ecological Monitoring Program in Korea.

**Figure 2 ijerph-17-02116-f002:**
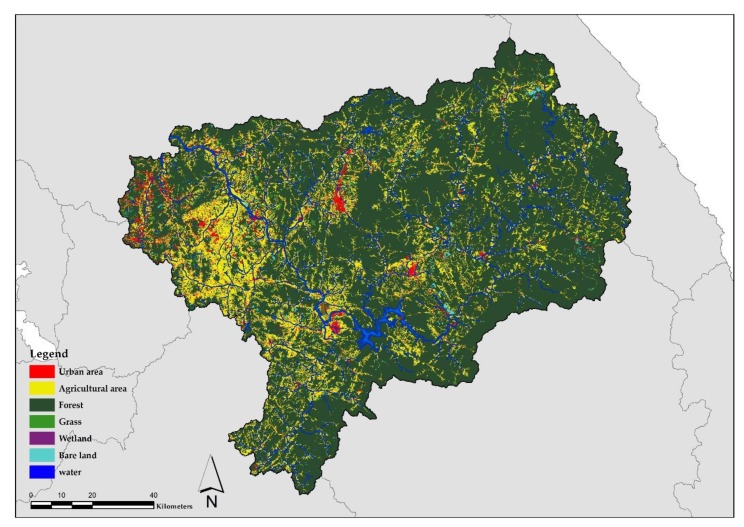
The spatial distribution of land use/land cover in the Nam-Han River watershed.

**Figure 3 ijerph-17-02116-f003:**
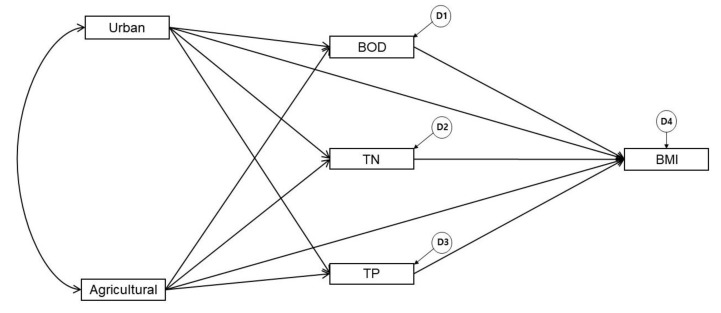
The hypnotized SEM describing the causal relationship between land use, water quality, and BMI.

**Figure 4 ijerph-17-02116-f004:**
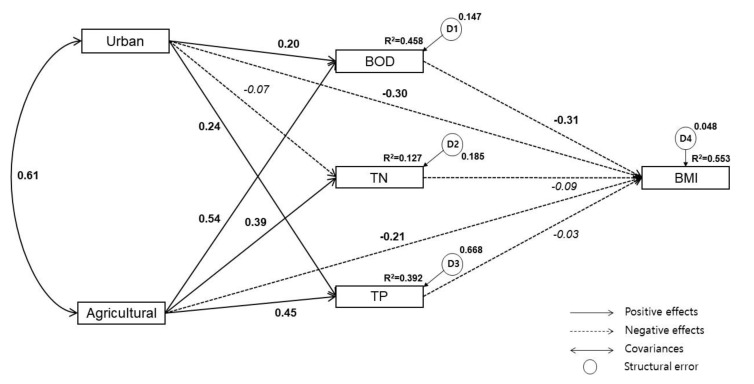
Initial SEM based on the hypothesis. The solid lines indicate positive effects, and the dotted lines indicate negative effects. Italicized values are statistically insignificant (*p* > 0.05).

**Figure 5 ijerph-17-02116-f005:**
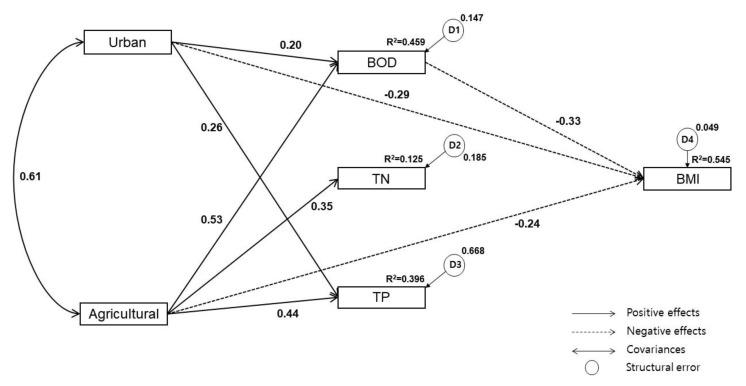
The refined model describing the causal relationships between water quality, land use, and BMI following the removal of insignificant pathways.

**Table 1 ijerph-17-02116-t001:** Descriptive statistics of water quality parameters, land use percent and BMI.

Classification	Variables	Min	Max	Mean	Std.*D*.
Water Quality Parameter	BOD (mg/L)	0.50	7.60	1.81	1.05
	T-N (mg/L)	0.69	11.20	3.85	1.88
	T-P (mg/L)	0.00	0.39	0.05	0.07
Land Use	Urban area (%)	0.00	21.57	5.87	4.58
	Agricultural area (%)	5.68	62.50	26.75	14.26
Biological Index	BMI (0~100)	22.6	94.7	66.71	18.82

n = 111. Std.*D*. = Standard Deviation, Min. = Minimum, Max. = Maximum.

**Table 2 ijerph-17-02116-t002:** Summary of the model fit indexes. All indexes infer high suitability of the estimated model to explain the data in this study.

Model Fit Index	Criteria	Estimated Model	Refined Model
NFI	≥0.90	0.909	0.992
TLI	≥0.90	0.575	1.012
CFI	≥0.90	0.915	1.000
GFI	≥0.90	0.921	0.993
AGFI	≥0.90	0.445	0.952
RMSEA	≤0.05	0.273	0.000

**Table 3 ijerph-17-02116-t003:** Relationships between urban and agricultural land use, water quality and BMI.

Path			Standardized Coefficients	S.E.	C.R.	*p*
Agricultural	→	BOD	0.53	0.08	6.064	0.001
Agricultural	→	TP	0.43	0.17	4.770	0.001
Agricultural	→	TN	0.35	0.07	3.957	0.001
Agricultural	→	BMI	−0.23	0.05	−2.557	0.011
Urban	→	BOD	0.20	0.04	2.329	0.020
Urban	→	TP	0.26	0.09	2.994	0.003
Urban	→	BMI	−0.29	0.02	−3.510	0.001
BOD	→	BMI	−0.33	0.05	−3.809	0.001

S.E. = standard error; C.R. = critical ratio.

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
