# Peer review of "An Estimated Structural Equation Model to Assess the Effects of Land Use on Water Quality and Benthic Macroinvertebrates in Streams of the Nam-Han River System, South Korea"

_ijerph, 2020, doi:10.3390/ijerph17062116_

Round 1

Reviewer 1 Report

Thank you for providing the opportunity to review the article “An Estimate Structural Equation…..of the Nam-Han River”. Overall, I thought the manuscript well written and the study well-designed. I appreciate the application of proper statistical approaches to analyze the collected data. While I approve of the manuscript’s structure and study design, I’m not convinced of the utility and practically of the statistical model platform (SEM) as it was applied here. How exactly would watershed managers use this model for decision making? Your refined results in Figure 5 show a direct impact between agriculture, urban land use and BMI. So what exactly does this mean? That change in percentages of agriculture and urban land cover will impact macoinvertebrates? In reality, it’s not that land cover has a direct impact on BMI but rather watershed biogeochemistry has a direct impact on BMI. While the SEM statistical model structure appears impressive I’m not sure of the overall utility of this model from a practical standpoint. The authors need to provide a greater argument for its application. Otherwise, it’s a theoretical exercise of statistical modeling with no application value. Also, it’s not completely clear how landuse proportion is being compared/related to a concentration (BOD, TP, TN) and a BMI. Are you relating landuse percentage to average concentration and BMI on a subwatershed basis?

Specific Points

Page 2, lines 50-58: Check the wording of this section. There are several “tense” issues, e.g. have vs. had, and run-on sentences. 

Page 2, line 58: Change “simplifies” to “changes”

Page 4, line 159: What do you mean by “legalized”? Would say “applied”

Page 8, line 273: what is “Ia” and “II”?

Figure 4: Make the arrow heads bigger in all figures so the reader knows which direction

Figure 5: What are “D1” D2 and D3?

Author Response

Reviewer 1

Thank you for providing the opportunity to review the article “An Estimate Structural Equation…..of the Nam-Han River”. Overall, I thought the manuscript well written and the study well-designed. I appreciate the application of proper statistical approaches to analyze the collected data.

>> Thank you for your time to review the manuscript. Your comments and suggestions helped us a lot for improving the revised manuscript.

While I approve of the manuscript’s structure and study design, I’m not convinced of the utility and practically of the statistical model platform (SEM) as it was applied here. How exactly would watershed managers use this model for decision making? Your refined results in Figure 5 show a direct impact between agriculture, urban land use and BMI. So what exactly does this mean? That change in percentages of agriculture and urban land cover will impact macoinvertebrates? In reality, it’s not that land cover has a direct impact on BMI but rather watershed biogeochemistry has a direct impact on BMI. While the SEM statistical model structure appears impressive I’m not sure of the overall utility of this model from a practical standpoint. The authors need to provide a greater argument for its application. Otherwise, it’s a theoretical exercise of statistical modeling with no application value.

>> Thank you for your critical comment. Considering the complex interactions among watershed characteristics, stream environments and biological communities, the direct impacts of land uses are difficult to identify from indirect impacts. Practically, the direct impacts of urban and agricultural land uses might be associated with reducing or degrading habitat size and quality (e.g., frequent construction projects), resulting in a decreasing population size and varying species composition of benthic macroinvertebrates when developing urban and agricultural areas. This is discussed at the end of Discussion Section (see L. 397-412).

Also, it’s not completely clear how landuse proportion is being compared/related to a concentration (BOD, TP, TN) and a BMI. Are you relating landuse percentage to average concentration and BMI on a subwatershed basis?

>> We used a sub-watershed scale, and it is specified at L. 196-197 in revised manuscript.

Specific Points

Page 2, lines 50-58: Check the wording of this section. There are several “tense” issues, e.g.

have vs. had, and run-on sentences. 

>> We corrected “tense” issue (see L. 59-68).

Page 2, line 58: Change “simplifies” to “changes”

>> We changed “simplifies” to “changes” (see L.72).

Page 4, line 159: What do you mean by “legalized”? Would say “applied”

>> We changed “legalized” to “applied” (see L.158).

Page 8, line 273: what is “Ia” and “II”?

>> “Ia” code is a unique code assigned by Korean Ministry of Environment water quality classes.

Figure 4: Make the arrow heads bigger in all figures so the reader knows which direction

>> Thank you for the comment. We changed the arrowhead in Figure 3, 4, and 5.

Figure 5: What are “D1” D2 and D3?

>> D1, D2, D3, and D3 are not measured structural errors of variables, and they are explained at L. 213-216.

Again, we appreciate your valuable comments and suggestions.

Reviewer 2 Report

General Comments

This article is a very nice contribution to the field of water quality modelling and is dealing with a very interesting ecological subject. The manuscript is well written and presents the results clearly. I have observed some very minor issues that need to be resolved, in order the manuscript to be suitable for publication.

Specific Comments

Line  102: please replace the ML abbreviation  for  maximum likelihood estimation with the term MLE

 Line 155: please use  chlorophyll-a instead of chlorophyll a

Line 173: The BMI is abbreviated previously, please use only the BMI term

Line 278: please move the n=111 to line 276-277 (Table 1 legend)

Line 323: please use the abbreviations under Table 3 into the Table legend

Author Response

This article is a very nice contribution to the field of water quality modelling and is dealing with a very interesting ecological subject. The manuscript is well written and presents the results clearly. I have observed some very minor issues that need to be resolved, in order the manuscript to be suitable for publication.

>> Thank you for your time to review the manuscript. Your comments and suggestions helped us a lot for improving the revised manuscript.

Specific Comments

Line 102: please replace the ML abbreviation for maximum likelihood estimation with the term MLE

>> Thank you. We replaced the term “ML” by “MLE” (L. 87).

Line 155: please use chlorophyll-a instead of chlorophyll a

>> Thank you. We replaced the term “chlorophyll a” by “chlorophyll-a.”

Line 173: The BMI is abbreviated previously, please use only the BMI term

>> Thank you. We corrected it as you suggested.

Line 278: please move the n=111 to line 276-277 (Table 1 legend)

>> According to the standard form of the journal, the proper location of “n=111” is the bottom of the table.

Line 323: please use the abbreviations under Table 3 into the Table legend

>> According to the standard form of the journal, the proper location of denotes is the bottom of the table.

Again, we appreciate your valuable comments and suggestions.

Reviewer 3 Report

The manuscript presents a very interesting study on the effects of land use/cover on stream water quality, considering water chemistry and macroinvertebrates. They have a large database with 111 sampling sites in a watershed in Korea, allowing to test direct and indirect effects of land use/cover and water chemistry on a biological index. There are some minor comments and questions to improve the text that follow below.

However, the study could be much more general if the authors could include a variable (Habitat and Riparian Index - HRI), that may help explain the patterns found for the macroinvertebrates. A detailed comment is included at the end of this review. Anyway, this strategy could allow to test simultaneously the effects of habitat effects at two spatial scales (land use/cover vs local riparian), that may influence macroinvertebrate communities in stream systems.

Lines 59-65 describes an example on fishes, although the previous phrase starts a discussion on macroinvertebrates. The authors should provide a discussion on macroinvertebrates.

Lines 73-74: the phrase is inverted: “the influx of pollutants in urbanization and concentrated agriculture is adversely affected by the water quality and the biological community structure”. I think the water quality and the biological community structure are affected by the pollutants, not the contrary.

Lines 74-77: The phrase is confusing.

Lines 77-78: what is the relationship between stream shape complexity and the present study?

Line 79: I don’t think there is little research on the relationship between land use, water quality and organisms. There are several studies around the world. Maybe the authors meant that in their study region there is little research; or maybe few using the SEM approach.

Lines 84-103: too much detail is given. SEM is a recognized statistical framework, and the authors should summarize how this approach could contribute to a better understanding of stream macroinvertebrates, using the studies already published as an example. Focus on the biological issues, not on statistical issues.

Lines 104-114: considering the above comment, how SEM would contribute to a better understanding of your system? Which predictions and hypotheses could be proposed?

Line 117: “was conducted on streams of the Han River”

Line 146: “ecological and environmental indices”

Lines 158-159: did not understand “BOD is generally legalized in Korea”

Lines 264-265: which test was used? This text belongs to the Methods section.

Which software was used in the SEM analyses?

Lines 265-267: did you log-transform percentages? Since you only have 5 variables, it would be useful to know which of them were log-transformed. Percentages are generally transformed using angular transformations t obtain normality.

Lines 282-301: give R² values for the endogenous variables, for both models.

Lines 284-292: the text gives the same information as Figure 4, and also repeats the coefficient values. Please rewrite focusing on the biological information.

Figures 4 and 5: values for D1, D2, D3 and D4 should be given.

Lines 305-307 repeat exactly the same information presented in Table 2 and should be rewritten.

Lines 313-322: the authors did not explain in the Methods what is the Critical Ratio and what it is used for. The values of the coefficients and CRs are repeated in Table 3. They should rewrite this paragraph focusing again on the biological information.

Table 3: what is standard regression weight? Are these the standardized coefficients? And regression weight? Are they the raw coefficients? There is too much repetition of values (e.g., coefficients, etc.) in the manuscript, sometimes the same information appears three times (in the Figure, in the Table, and in the text). The authors should reconsider the presentation of their results.

Lines 340-342: what is the definition of “fields”? This phrase is confusing.

Lines 359-357: in fact, you have no data on sedimentation to propose this discussion, so it is speculation. See the next comment.

Lines 403-412: Here you have a more plausible explanation. Other studies also found that stream water nutrients were not related with macroinvertebrate indicators. For example, Tanaka et al. (2016, Agriculture, Ecosystems and Environment 216: 333-339) found that macroinvertebrate indicators were more related to dissolved oxygen concentrations (as in the present study) and to riparian forest structure, suggesting a more local response of the organisms, as also suggested in your discussion. However, I think you have data to test directly this effect: in lines 147-148 you informed that the MOE has data on Habitat and Riparian Index (HRI) and the Riparian Vegetation Index (RVI). Apparently, the HRI could be used as an endogenous variable in the SEM model, in the sequence: land use -> HRI -> water chemistry -> BMI. The complete model should be evaluated. In this way, the effects of drivers at different scales on water chemistry and BMI could be evaluated. Is HRI had an influence on BMI via water chemistry, this could strengthen the argument on the effects of sedimentation on macroinvertebrate communities. If HRI had a direct effect on BMI, would it be stronger or weaker than land use/cover effects? Several studies suggest that macroinvertebrates respond more to local drivers. With this analysis, the study could be far more interesting, by incorporating a variable at a local scale so that effects at different spatial scales could be evaluated.

Lines 404-406: I think the phrase is reversed. For example, shrubs of various widths remove 66.4% of the phosphorus, not the contrary. A text review would be welcome.

Author Response

The manuscript presents a very interesting study on the effects of land use/cover on stream water quality, considering water chemistry and macroinvertebrates. They have a large database with 111 sampling sites in a watershed in Korea, allowing to test direct and indirect effects of land use/cover and water chemistry on a biological index. There are some minor comments and questions to improve the text that follow below.

 >> Thank you for your valuable comments and encouragement.

However, the study could be much more general if the authors could include a variable (Habitat and Riparian Index - HRI), that may help explain the patterns found for the macroinvertebrates. A detailed comment is included at the end of this review. Anyway, this strategy could allow to test simultaneously the effects of habitat effects at two spatial scales (land use/cover vs local riparian), that may influence macroinvertebrate communities in stream systems.

>> Thank you for your interesting suggestions. As you suggested, we integrated HRI into our study SEM, and it turned the whole model was not statistically significant. Also, we found a low correlation between BMI and HRI. Then, we looked into variables used for computing HRI, separately. It was found that some variables of HRI were sensitive to our model. However, it was not easy to determine to use or not to use some variables of HRI, and we decided not to integrate HRI into our model in this study. In sum, we decided not to integrate HRI into our model in this study although your suggestion was very interesting. We hope you understand time limit given by the editorial office of the journal for resubmitting the revised manuscript.      

Lines 59-65 describes an example on fishes, although the previous phrase starts a discussion on macroinvertebrates. The authors should provide a discussion on macroinvertebrates.

 >> We revised this part focusing on macroinvertebrates with some findings of previous studies (L. 59-68).

Lines 73-74: the phrase is inverted: “the influx of pollutants in urbanization and concentrated agriculture is adversely affected by the water quality and the biological community structure”. I think the water quality and the biological community structure are affected by the pollutants, not the contrary.

>> It was falsely stated, and we corrected it (L. 59-60).

Lines 74-77: The phrase is confusing.

 >> We rephrased these sentences (L. 60-68).  

Lines 77-78: what is the relationship between stream shape complexity and the present study?

 >> We agreed. This line is removed. 

Line 79: I don’t think there is little research on the relationship between land use, water quality and organisms. There are several studies around the world. Maybe the authors meant that in their study region there is little research; or maybe few using the SEM approach.

>> You are right. We wanted to emphasized lack of systemic view of the relationships among land uses, water quality and biological indicators in streams, but it was not clear. We rephrased the entire paragraph (L. 90-97).

Lines 84-103: too much detail is given. SEM is a recognized statistical framework, and the authors should summarize how this approach could contribute to a better understanding of stream macroinvertebrates, using the studies already published as an example. Focus on the biological issues, not on statistical issues.

>> Agreed. We rephrased the paragraphs (L. 90-97, L. 104-112).    

Lines 104-114: considering the above comment, how SEM would contribute to a better understanding of your system? Which predictions and hypotheses could be proposed?

>> We specified the hypothesis at the end of Introduction (L. 104-107). 

Line 117: “was conducted on streams of the Han River”

 >> We change the sentence as “Our study area was the Nam-Han River watershed…” (L. 115).

Line 146: “ecological and environmental indices”

>> We change the sentence (L.145).

Lines 158-159: did not understand “BOD is generally legalized in Korea”

 >> We change the sentence as “BOD is generally applied in Korea and is useful..”(L.158).

Lines 264-265: which test was used? This text belongs to the Methods section.

  >> Kolmogorov-Smirnov test was conducted, and it was moved to Method section (L. 205).

Which software was used in the SEM analyses?

 >> We used AMOS 18 of SPSS, and it is specified at L. 205.

Lines 265-267: did you log-transform percentages? Since you only have 5 variables, it would be useful to know which of them were log-transformed. Percentages are generally transformed using angular transformations t obtain normality.

 >> All variables were log-transformed, and it is explained at L. 268.

Lines 282-301: give R² values for the endogenous variables, for both models.

 >> We provided R2 values in Figure 4&5.

Lines 284-292: the text gives the same information as Figure 4, and also repeats the coefficient values. Please rewrite focusing on the biological information.

>> We tried not to repeat the number shown in Figures in the revised manuscript.

Figures 4 and 5: values for D1, D2, D3 and D4 should be given.

 >> We provided values of D1-D4 in Figure 4&5.

Lines 305-307 repeat exactly the same information presented in Table 2 and should be rewritten.

>> We removed repeated sentence with the table from the manuscript.

Lines 313-322: the authors did not explain in the Methods what is the Critical Ratio and what it is used for. The values of the coefficients and CRs are repeated in Table 3. They should rewrite this paragraph focusing again on the biological information.

>> The term C.R. is explained in Method section (L. 231), and we removed the repeated information. 

Table 3: what is standard regression weight? Are these the standardized coefficients? And regression weight? Are they the raw coefficients? There is too much repetition of values (e.g., coefficients, etc.) in the manuscript, sometimes the same information appears three times (in the Figure, in the Table, and in the text). The authors should reconsider the presentation of their results.

 >> The term “standard regression weight” refers to “standardized coefficient” in general regression. For avoiding unnecessary confusion, we decided to use the term “standardized coefficient.”

Lines 340-342: what is the definition of “fields”? This phrase is confusing.

>> We replace “field” by “areas.”

Lines 359-357: in fact, you have no data on sedimentation to propose this discussion, so it is speculation. See the next comment.

>> Agreed. We did not integrate sediment into our model. However, numerous previous studies reported that nutrients discharge is closely tied with sediment load and deposition. Thus, we believed that discussing nutrient concentration with sediment is fair approach. Nonetheless, the entire paragraph was rephrased for clarity (L. 359-379.          

Lines 403-412: Here you have a more plausible explanation. Other studies also found that stream water nutrients were not related with macroinvertebrate indicators. For example, Tanaka et al. (2016, Agriculture, Ecosystems and Environment 216: 333-339) found that macroinvertebrate indicators were more related to dissolved oxygen concentrations (as in the present study) and to riparian forest structure, suggesting a more local response of the organisms, as also suggested in your discussion. However, I think you have data to test directly this effect: in lines 147-148 you informed that the MOE has data on Habitat and Riparian Index (HRI) and the Riparian Vegetation Index (RVI). Apparently, the HRI could be used as an endogenous variable in the SEM model, in the sequence: land use -> HRI -> water chemistry -> BMI. The complete model should be evaluated. In this way, the effects of drivers at different scales on water chemistry and BMI could be evaluated. Is HRI had an influence on BMI via water chemistry, this could strengthen the argument on the effects of sedimentation on macroinvertebrate communities. If HRI had a direct effect on BMI, would it be stronger or weaker than land use/cover effects? Several studies suggest that macroinvertebrates respond more to local drivers. With this analysis, the study could be far more interesting, by incorporating a variable at a local scale so that effects at different spatial scales could be evaluated.

>> As you previously suggested, building a model as in land use -> HRI -> water chemistry -> BMI sequence is very interesting and ideal. However, our estimation with HRI appeared to be not significant (please see above). This ideal model can be studied with multiple control variables.    

 Lines 404-406: I think the phrase is reversed. For example, shrubs of various widths remove 66.4% of the phosphorus, not the contrary. A text review would be welcome.

>> You are right. Nonetheless, the entire paragraph was rephrased (L. 400-415).

We appreciated your critical comments and suggestions.